# Spleen Swabs for Sensitive and High-Throughput Detection of African Swine Fever Virus by Real-Time PCR

**DOI:** 10.3390/v16081316

**Published:** 2024-08-18

**Authors:** Christopher Cafariello, Kalhari Goonewardene, Chungwon J. Chung, Aruna Ambagala

**Affiliations:** 1National Centre for Foreign Animal Disease, Canadian Food Inspection Agency, Winnipeg, MB R3E 3M4, Canada; chris.cafariello@inspection.gc.ca (C.C.); kalhari.goonewardene@inspection.gc.ca (K.G.); 2Foreign Animal Disease Diagnostic Laboratory, Animal and Plant Health Inspection Services, United States Department of Agriculture, Plum Island Animal Disease Center, Greenport, NY 11944, USA; chungwon.chung@usda.gov; 3Department of Medical Microbiology and Infectious Diseases, Max Rady College of Medicine, University of Manitoba, Winnipeg, MB R3E 0J9, Canada; 4Comparative Biology and Experimental Medicine, Faculty of Veterinary Medicine, University of Calgary, Calgary, AB T2N 1N4, Canada

**Keywords:** African swine fever, diagnostics, tissue homogenate, spleen swab, real-time PCR

## Abstract

African swine fever (ASF) continues to spread in Africa, Europe, Asia and the island of Hispaniola, increasing the need to develop more streamlined and highly efficient surveillance and diagnostic capabilities. One way to achieve this is by further optimization of already established standard operating procedures to remove bottlenecks for high-throughput screening. Real-time polymerase chain reaction (real-time PCR) is the most sensitive and specific assay available for the early detection of the ASF virus (ASFV) genome, but it requires high-quality nucleic acid extracted from the samples. Whole blood from live pigs and spleen tissue from dead pigs are the preferred samples for real-time PCR. Whole blood can be used as is in nucleic acid extractions, but spleen tissues require an additional homogenization step. In this study, we compared the homogenates and swabs prepared from 52 spleen samples collected from pigs experimentally inoculated with highly and moderately virulent ASF virus strains. The results show that not only are the spleen swabs more sensitive when executed with a low-cell-count nucleic acid extraction procedure followed by real-time PCR assays but they also increase the ability to isolate ASFV from positive spleen samples. Swabbing is a convenient, simpler and less time-consuming alternative to tissue homogenization. Hence, we recommend spleen swabs over tissue homogenates for high-throughput detection of ASFV by real-time PCR.

## 1. Introduction

African swine fever (ASF) is a contagious, lethal hemorrhagic fever that affects both wild and domestic swine, leading to huge economic losses in the swine industry as well as increased food insecurity [1,2]. The causative agent, ASF virus (ASFV), is an enveloped, double-stranded DNA arbovirus now endemic to Sub-Saharan Africa, much of Eastern Europe, Asia and the Central American island of Hispaniola [3,4,5,6,7,8,9]. As the virus spreads across the globe, many countries have increased their surveillance efforts for early detection of ASFV genomic DNA [10,11]. 

Real-time PCR is the most sensitive and specific assay available for the detection of ASFV genomic DNA in clinical samples. It is both highly scalable and can be easily automated, making it ideal for use as a screening assay in most diagnostic laboratories. The performance of real-time PCR assays relies on the quality of the nucleic acid extracted from samples, which in turn depends on the sample preparation methodology used. Whole blood from ASF suspected live animals as well as spleen tissues from dead animals are the preferred World Organization for Animal Health (WOAH)-recommended sample types for ASFV genome detection [12]. Spleen tissue is highly enriched for ASFV particles in affected swine and is therefore used in diagnostic and surveillance activities in animal health laboratories in many countries [13,14]. However, unlike whole blood, which can be used without further processing, spleen samples need to be processed to generate 10% (*w*/*v*) homogenates in sterile PBS. Preparation of 10% homogenates involves cutting, weighing and the process of homogenization, which is both time-consuming and labor-intensive, reducing the throughput and delaying results. Traditionally, homogenization was carried out using a mortar and pestle. A pestle is a hard, blunt object which is used to grind the sample against a mortar, which is a solid container. This is a low-throughput method because only one sample can be processed at a time. The tissue homogenizers have increased the throughput of the homogenization step; however, they are expensive and therefore not accessible to countries with limited resources. Swabbing spleen tissues is a straightforward alternative to tissue homogenization because it directly eliminates the weighing and homogenization steps. 

In this study, we compared the use of traditional 10% spleen tissue homogenates to spleen swabs for nucleic acid extraction and ASFV genome detection by real-time PCR. Spleen tissues collected from pigs experimentally infected with three different ASFV strains along with healthy ASF known-negative domestic and wild pigs were used in this study.

## 2. Materials and Methods

### 2.1. Sample Origins

Spleen samples used in this study were collected from pigs experimentally inoculated with ASFV p72 genotype I (ASFV Malta’78) and II (ASFV Estonia/2014 and ASFV Georgia 2007) viruses at different days post infection (dpi). ASFV Malta’78 is a naturally attenuated moderately virulent strain isolated from Malta in 1978 [15], which was kindly provided by Dr. Linda Dixon from the Pirbright Institute, UK. ASFV Estonia/2014 is a naturally attenuated, moderately virulent strain isolated in 2014 from Estonia [16,17], which was kindly provided by Dr. Sandra Blome from the FLI, Germany. ASFV Georgia 2007 is a highly virulent strain isolated in 2007 from Georgia [18]. All animal experiments were conducted at the biosafety level 3 facility at the NCFAD, Winnipeg, MB, Canada, under the guidelines of the Canadian Council for Animal Care. The Animal Care Committee at the Canadian Science Centre for Human and Animal Health approved the animal use under the AUDs C-19-012 and C-22-003. The samples were used immediately after collection or were frozen at −80 °C. Before use, the frozen spleen samples were thawed at room temperature.

### 2.2. Preparation of Homogenates and Swabs from Spleen Tissue Samples

For homogenization of spleen tissue, a Precellys^®^ 24 Touch instrument (Bertin technologies, Rockville, MD, USA) was used. Spleen tissue was cut into small pieces using sterile scissors or a scalpel. Approximately 0.1 g of spleen tissue was weighed using a small laboratory scale before being transferred to homogenization tubes, each containing 1 mL of sterile PBS. The tubes were then closed tightly and homogenized twice at 3 × 10 s at 5000 RPM. The homogenate was then transferred to fresh cryovials, spun at 2000× *g*, for 20 min at 4 °C and 55 µL of supernatant from each tube was used for nucleic acid extraction. For the nucleic acid extraction, a MagMax™ Pathogen RNA/DNA kit (Thermo Fisher Scientific, Waltham, MA, USA) was used according to the standard protocol provided by the manufacturer on the MagMax™ KingFisher Apex Deep Well Magnetic Particle Processor (Thermo Fisher Scientific).

Spleen swabs were collected using sterile polyester-tipped applicator swabs (Puritan #25-806-1PD, Puritan Medical Products, Falmouth, ME, USA). The spleen was cut open with a sterile scalpel and the cotton swab was inserted into the incision, pressed firmly and twisted until the tip was fully soaked (Appendix A). Cutting the spleen was not always necessary since some of the spleen tissues were fragile and the swabs could be inserted directly into the spleen by simply pressing the swab firmly against the tissue. The soaked ends of the swabs were then submerged into cryovials each containing 1 mL of sterile PBS. The excess swab length was snapped off, the cryovials were closed and vortexed for at least 10 s and 200 µL of the liquid from each vial was used for nucleic acid extraction, according to the low-cell-count optimized protocol provided by the manufacturer. Following the initial nucleic acid extractions, the remaining samples were stored at −70 °C for further testing.

### 2.3. Real-Time PCR Detection of ASFV Genomic DNA, β-Actin and Armored Enterovirus RNA

ASFV genomic DNA in both homogenate and swab samples was detected using two quantitative real-time PCR assays, both targeting the highly conserved region of the ASFV p72 open reading frame. As reported previously, the limit of detection of the Tignon assay is between 5.7 and 57 copies of the ASFV genome while the sensitivity of the Zsak assay is between 1.4 and 8.4 ASFV genome copies [19,20]. A separate real-time PCR assay specific for β-actin (Moniwa assay) was included to determine efficient nucleic acid extraction and amplification [21]. For testing for the presence of PCR inhibitors, an armored enterovirus RNA spike-in (Cat #42050, Assurgent, Inc., Austin, TX, USA) was used. All real-time PCR assays were run as singleplex assays and the reactions were prepared using TaqMan™ Fast Virus 1-Step Master Mix (Thermo Fisher Scientific) and were amplified using the Bio-Rad CFX96 instrument (Bio-Rad, Mississauga, ON, Canada), using the recommended cycling conditions (50 °C for 5 min, 95 °C for 20 s, followed by 95 °C for 3 s and 60 °C for 30 s) for the TaqMan™ Fast Virus 1-Step Master Mix. To determine C_t_ values, a positive control well was included, while a regression analysis of the positive control alone was used to determine a threshold cutoff for all experimental wells [22].

### 2.4. Virus Isolation

Virus isolation was carried out in porcine primary leukocytes (PPLs), following the NCFAD standard operating procedure for ASFV isolation [23]. Briefly, PPLs were isolated from fresh porcine blood and plated in 24-well plates, 1 mL/well (10^6^ WBC/mL with 0.4% *v*/*v* RBC). Forty-eight hours later, 11 selected paired homogenate and swab samples along with a no-virus control (#8, 10, 12–14, 41–46) were inoculated (20 µL per well) into PPL cultures. Cultures were then incubated at 37 °C in a 5% CO_2_ incubator for 7 days and observed daily for the appearance of hemadsorption (HAD). The appearance of HAD was considered a positive indication of isolation. The limit of detection of the NCFAD ASFV isolation protocol is approximately 10^2^ tissue culture infectious dose (TCID)_50_/mL.

### 2.5. Virus Titration

A selected number of homogenates and swab samples (#11, 18, 38) were titered on primary porcine alveolar macrophage (PAM) cultures, as previously described [24]. Briefly, 10-fold dilutions of the isolated viruses were inoculated into 90% confluent primary PAM cells in α-MEM supplemented with 1% Gentamicin, 1% Glutamax and 2% γ-irradiated fetal bovine serum. Following 3 days of incubation at 37 °C and 5% CO_2_, the plates were fixed and stained with an anti-ASFV polyclonal pig serum and HRP-conjugated goat anti-pig monoclonal antibody (Cat# 114-035-003; Jackson ImmunoResearch, West Grove, PA, USA).

### 2.6. Statistical Analysis

The Pearson correlation coefficient between the C_t_ values for the homogenate and the swab samples was calculated using Graph Pad Prism Software, version 9.0.2 (San Diego, CA, USA).

## 3. Results and Discussion

In this study, 52 spleen samples collected from pigs experimentally infected with highly and moderately virulent ASFV strains and 20 known-negative spleen samples (14 wild boar and 6 domestic pig spleen samples) collected under the ongoing CanSpot Canadian ASF surveillance system (https://www.animalhealthcanada.ca/canspotasf, accessed on 15 June 2024) were processed in parallel into both 10% (*w*/*v*) tissue homogenates and spleen swabs. For nucleic acid extraction, an input volume of 55 µL was maintained for the homogenates while a low-cell-count protocol with an input volume of 200 µL was used for swabs. All samples were tested for the presence of ASFV genomic DNA using both the Tignon and Zsak real-time PCR assays, and for pig ß-actin (internal sample control) using the Moniwa real-time PCR assay.

ß-actin was detected in all homogenates, as well as in all spleen swab samples, suggesting comparable levels of splenic tissue sequestered by the swabs. A similar trend was observed with ASFV genomic detection (Table 1).

The Tignon assay was able to detect ASFV genomic DNA in 39 out of 52 homogenates (75.0% positivity) prepared from spleen samples collected from pigs experimentally inoculated with ASFV. The same assay was able to detect ASFV genomic DNA in 45/52 (86.5%) of the swab samples collected from the same spleen samples.

The Zsak assay was able to detect ASFV genomic DNA in 40/52 (76.9%) homogenates and 46/52 (88.5%) swabs. A positive correlation (Figure 1) was observed between the homogenates and the swabs in all three assays (Tignon = 0.96, Zsak = 0.97 and ß-actin = 0.71).

ASFV genomic DNA was not detected in any of the known-negative samples from the 14 wild boar and 6 domestic pig spleen samples by both the Tignon and Zsak assays while ß-actin was detected in all samples, demonstrating that both ASFV assays are specific (Appendix A).

Both the Tignon and Zsak assays failed to detect ASFV genomic DNA in six homogenate samples compared to their paired spleen swab samples. Those swab samples (#6, 10, 27, 28, 30 and 31 for Tignon, and # 10, 27, 28, 30, 31 and 33 for Zsak) had very low ASFV genome levels and therefore the discrepancy could be due to the difference in input sample volumes (55 µL for homogenates and 200 µL for swabs) used for nucleic acid extractions. To address this, 10 paired homogenate and swab samples were selected, and nucleic acid was extracted from homogenate samples using the low-cell-count nucleic acid extraction protocol which uses 200 µL input. Despite the same input volume being used, swabs remained more sensitive than homogenates (Table 2). Both Tignon and Zsak assays failed to detect ASFV genomic DNA in the homogenate from sample 27. Likewise, the Tignon assay failed to detect in the homogenate from sample 28 while the Zsak assay in the homogenates from samples 30 and 31.

The decreased sensitivity observed with homogenates could also be due to the presence of PCR inhibitors in extracted nucleic acids [25]. The presence of PCR inhibitors is the most common reason for poor PCR performance when DNA yield is sufficient. PCR inhibitors can derive either from the original sample or from sample preparation prior to PCR [26]. The possible effects of PCR inhibitors in nucleic acid extracted from homogenates vs. swabs were evaluated by spiking the nucleic acid extracts from a selected number of homogenates and swab samples with armored enterovirus RNA and real-time PCR (Appendix A). The C_t_ values from the armored enterovirus RNA real-time PCR for homogenates and spleen swabs were comparable.

Another potential contributor to the observed increase in sensitivity in swabs compared to homogenates could be that the swabs are able to better capture and concentrate the splenic cells including red blood cells with high affinity to ASFV [27]. To determine if ASFV-positive spleen swabs contain comparable or slightly higher levels of intact ASFV particles, a selected number of ASFV-genome-positive spleen swabs and paired tissue homogenate samples were inoculated into primary PPL cultures. ASFV was isolated from both sample types; however, hemadsorption (HAD) was clearer and more abundant in PPL cultures inoculated with spleen swab samples compared to those inoculated with the paired homogenate samples. HAD was observed in 4 out of 11 PPL cultures inoculated with swab samples compared to 3 out of 11 PPL cultures inoculated with homogenate samples (Appendix A). Ten-fold dilutions of three spleen swab samples with variable C_t_ values and paired homogenate samples (Table 3) were also inoculated onto PAM cells to determine the ASFV titers for each sample type. Consistent with the real-time PCR and ASFV isolation data, spleen swabs had slightly higher viral titers than homogenates.

Collectively, the results from this study show that the spleen swab samples provide comparable, if not slightly increased, sensitivity compared to the spleen tissue homogenates routinely used for ASFV genomic detection. Swabbing spleen tissue requires less hands-on time and offers a straightforward methodology for clinical sample preparation for ASFV genomic DNA detection. The exact mechanism for the increased sensitivity is not clear but could simply be that the swabs cover more splenic surface area than tissue homogenization.

Swabbing tissues as a sample preparation methodology for virus detection, has been explored for several other animal viruses. During an equine influenza outbreak in New South Wales, Australia, complications and delays associated with homogenizing tissue samples were avoided by directly swabbing freshly cut tissues [28]. This approach along with the incorporation of an influenza-specific real-time PCR assay helped laboratories to dramatically increase the number of samples screened during the outbreak. In a relatively small study conducted by Errington et al., spleen swabs were tested for the detection of bovine viral diarrhea virus (BVDV) and porcine reproductive and respiratory syndrome virus (PRRSV) [27]. The results demonstrated that the spleen swabs were a useful alternative to the traditional tissue lysis approach for the detection of both viruses. In a more recent study, Okwasiimire et al. compared diaphragm meat juice and muscle swab samples to spleen and spleen swab samples for the detection of ASFV nucleic acid [29]. The samples were collected from pigs with signs of ASFV infection at slaughterhouses near Kampala, Uganda. In this study, a high correlation (r = 0.8728) was observed between spleen tissue and spleen swab samples. Out of 493 samples tested, the positivity in spleen swabs (48.9%) was slightly lower than for spleen tissues (50.1%). The difference could be due to many reasons including the type of swabs used, the increased volume of storage medium (2–3 mL versus 1 mL in our study) and the use of different nucleic acid extraction methodology (column-based vs magnetic based extraction) used in the study. Spleen swabs are not yet approved by the WOAH; however, the USDA approved the use of spleen swabs in the USA [30].

## 4. Conclusions

Early detection of an ASF incursion is critical to prevent the severe economic and social impacts related to loss of animals, loss of trade and re-attaining a disease-free status [31]. To achieve this, many nations have implemented passive surveillance programs. Whole blood from live pigs and spleen samples from dead pigs are the preferred WOAH-recommended sample types for ASFV genome detection by real-time PCR followed by virus isolation. While whole blood can be used directly in nucleic acid extraction protocols, current laboratory standard operating procedures for spleen tissue samples require processing into 10% homogenates, which is both time-consuming and requires additional resources, significantly reducing throughput. In this study, we have shown that the spleen swabs are a better alternative to spleen tissue homogenates for both ASFV genome detection and virus isolation. Increased efficiency in sample processing, preparation and testing is critical for expanding ongoing surveillance and maintaining diagnostic throughput in the event of an outbreak. The use of spleen swabs instead of spleen tissue homogenates will not only allow animal health laboratories to reduce the costs associated with reporting but also streamline and increase the diagnostic throughput for ASF detection.

## Figures and Tables

**Figure 1 viruses-16-01316-f001:**
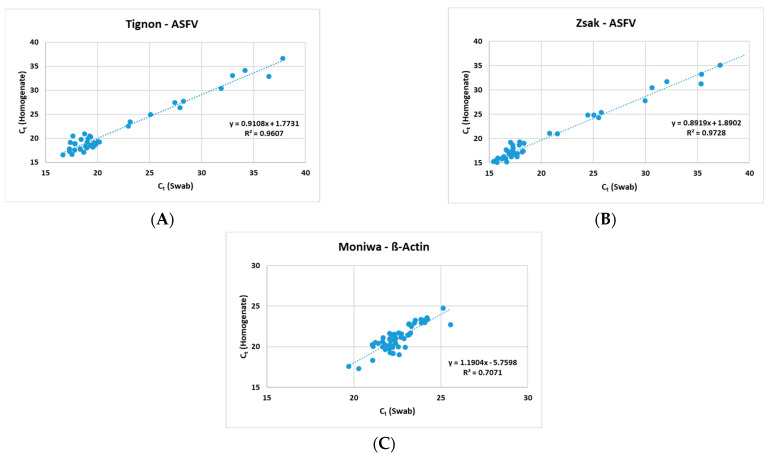
Positive correlation observed between 10% homogenates and spleen swabs by (**A**) Tignon ASFV, (**B**) Zsak ASFV and (**C**) Moniwa β-actin assays.

**Table 1 viruses-16-01316-t001:** Real-time PCR results for 52 spleen samples collected from pigs experimentally infected with ASFV. ND = not detected (highlighted in grey).

Strain	Sample	dpi	Tignon-ASFV		Zsak-ASFV		Moniwa-B-Actin	
			Homogenate	Swab	Homogenate	Swab	Homogenate	Swab
ASFV Malta’78	1	3	23.15	23.47	20.78	21.13	21.08	18.32
	2	4	19.53	18.21	17.26	17.08	21.66	20.64
	3	5	18.35	17.80	16.60	15.98	22.24	20.86
	4	3	28.28	27.74	25.74	25.43	21.40	20.44
	5	5	17.55	16.68	15.76	15.17	22.72	21.62
	6	2	ND	37.21	37.19	35.08	22.23	19.18
	7	4	20.05	19.39	18.25	17.42	22.13	21.47
	8	1	ND	ND	ND	ND	22.09	19.30
	9	4	18.83	18.36	16.39	16.29	22.10	20.72
	10	1	ND	39.88	ND	39.28	21.66	20.00
	11	3	27.92	26.40	25.52	24.32	21.11	20.04
	12	2	ND	ND	ND	ND	22.04	19.79
	13	1	ND	ND	ND	ND	21.07	20.25
	14	4	20.16	19.24	18.17	17.25	22.34	21.53
	15	3	27.43	27.50	25.04	24.88	22.97	19.95
ASFV Georgia 2007/1	16	18	36.46	32.95	35.36	31.24	19.70	17.57
	17	18	22.97	22.57	21.53	21.00	20.28	17.33
	18	17	16.66	16.58	15.85	16.01	22.05	21.67
	19	16	18.33	17.73	17.12	17.16	22.71	21.17
	20	6	17.64	20.48	17.06	19.20	23.19	22.85
	21	11	18.76	20.92	17.90	19.32	23.16	22.79
	22	18	19.32	20.36	18.33	19.08	23.32	22.53
	23	10	19.07	20.01	17.88	18.69	24.06	23.28
	24	18	17.83	18.89	16.76	17.45	23.26	21.69
	25	18	25.09	24.96	24.46	24.85	22.26	19.20
	26	22	17.37	19.14	16.63	17.73	23.87	23.34
	27	25	ND	35.21	ND	34.76	22.35	21.29
	28	25	ND	35.64	ND	33.74	21.68	21.12
	29	22	17.81	18.87	17.30	18.05	25.13	24.77
	30	25	ND	37.13	ND	36.25	21.22	20.53
	31	25	ND	37.54	ND	36.38	22.01	20.15
	32	25	18.44	19.81	17.27	18.66	25.57	22.72
ASFVE stonia/2014	33	5	ND	ND	ND	38.13	22.62	19.01
	34	11	ND	ND	ND	ND	22.55	20.00
	35	22	ND	ND	39.47	ND	22.06	20.93
	36	37	ND	ND	ND	ND	22.18	21.48
	37	6	34.18	34.13	32.05	31.71	22.17	20.16
	38	20	31.89	30.40	29.99	27.76	22.23	19.94
	39	26	37.80	36.66	35.39	33.22	22.42	20.40
	40	33	32.96	33.08	30.66	30.47	21.82	19.69
	41	1	19.27	20.48	17.25	17.63	21.79	20.25
	42	14	19.69	19.04	17.30	16.95	22.59	21.69
	43	30	18.68	17.14	16.65	15.23	23.90	22.93
	44	39	18.99	19.25	16.91	16.86	23.18	21.47
	45	2	19.82	18.85	17.63	16.62	22.41	21.03
	46	15	19.73	18.55	17.69	16.25	23.10	21.46
	47	27	19.39	18.61	17.50	16.95	22.87	21.03
	48	34	19.28	18.42	17.73	16.95	24.22	23.38
	49	3	17.81	17.57	16.19	15.84	23.48	22.94
	50	13	18.96	18.02	17.13	16.31	23.52	23.23
	51	29	17.29	17.85	15.70	15.66	24.07	22.98
	52	36	17.31	17.26	15.44	15.34	24.22	23.56
**Percent (%) Detection**			**75.0**	**86.5**	**76.9**	**88.5**	**100.0**	**100.0**

**Table 2 viruses-16-01316-t002:** Real-time PCR results of paired 10% homogenate and spleen swab samples using nucleic acids extracted using the low-cell-count nucleic acid extraction protocol which includes a 200 µL input volume. ND = not detected (highlighted in grey).

Strain	Sample	Tignon-ASFV		Zsak-ASFV		Moniwa-B-Actin	
		Homogenate	Swab	Homogenate	Swab	Homogenate	Swab
ASFV Georgia 2007/1	16	38.27	33.63	35.06	30.88	19.36	18.46
	17	22.41	22.62	21.34	20.66	19.91	18.22
	18	16.38	16.55	16.03	15.84	21.36	21.44
	26	17.97	18.88	17.02	17.73	23.07	22.83
	27	ND	34.07	ND	34.04	21.13	21.96
	28	ND	36.51	38.39	32.77	20.45	20.46
	29	19.02	18.88	18.00	17.92	24.24	24.07
	30	39.59	38.03	ND	34.56	19.98	20.46
	31	39.53	36.42	ND	37.16	20.52	21.04
	32	16.02	20.26	15.635	19.04	24.75	23.22
**Percent (%) Detection**		**80.0**	**100.0**	**70.0**	**100.0**	**100.0**	**100.0**

**Table 3 viruses-16-01316-t003:** ASFV titers in spleen swab samples vs. in paired tissue homogenates. TCID_50_ = tissue culture infectious dose 50%.

Sample	Tignon-ASF		Titer-Log (TCID_50_/mL)	
	Homogenate	Swab	Homogenate	Swab
11	27.92	26.40	4.10	4.87
18	16.66	16.58	7.53	8.65
38	31.89	30.40	3.97	4.80

## Data Availability

The data presented in this study are available on request from the corresponding author.

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
