# Peer review of "Spleen Swabs for Sensitive and High-Throughput Detection of African Swine Fever Virus by Real-Time PCR"

_viruses, 2024, doi:10.3390/v16081316_

Round 1
Reviewer 1 Report
Comments and Suggestions for Authors
This study aimed to enhance African swine fever virus (ASFV) detection efficiency by comparing traditional spleen tissue homogenates with spleen swabs. Using real-time PCR, it was found that spleen swabs were more sensitive and less labor-intensive than homogenates. The swabs also allowed for easier ASFV isolation from positive samples. Consequently, the study recommends spleen swabs as a superior alternative to homogenates for ASFV detection.
The paper is clearly written, making the main arguments straightforward and easy to understand. The discussion section adequately addresses any questions that arose during reading, providing a comfortable and informative experience. With only minor revisions, the paper should be ready for publication in the intended journal.
- To avoid confusion between real-time PCR and reverse transcriptase PCR, please replace all abbreviations of "RT-PCR" with "real-time PCR" in the manuscript. If there are any references to reverse transcriptase PCR genuinely intended as "RT-PCR," those can remain as is.
- In line 168, please change "40 (76.9%)" to "40/52 (76.9%)", and in line 169, change "46 (88.5%)" to "46/52 (88.5%)".
Author Response
Reviewer #1:
Comments
This study aimed to enhance African swine fever virus (ASFV) detection efficiency by comparing traditional spleen tissue homogenates with spleen swabs. Using real-time PCR, it was found that spleen swabs were more sensitive and less labor-intensive than homogenates. The swabs also allowed for easier ASFV isolation from positive samples. Consequently, the study recommends spleen swabs as a superior alternative to homogenates for ASFV detection.
The paper is clearly written, making the main arguments straightforward and easy to understand. The discussion section adequately addresses any questions that arose during reading, providing a comfortable and informative experience. With only minor revisions, the paper should be ready for publication in the intended journal.
- To avoid confusion between real-time PCR and reverse transcriptase PCR, please replace all abbreviations of "RT-PCR" with "real-time PCR" in the manuscript. If there are any references to reverse transcriptase PCR genuinely intended as "RT-PCR," those can remain as is.
- In line 168, please change "40 (76.9%)" to "40/52 (76.9%)", and in line 169, change "46 (88.5%)" to "46/52 (88.5%)".
Response:
We thank the reviewer for all the good points. Please see the minor edits made in the manuscript following the suggestions.
All “RT-PCR” was changed to “real-time PCR”.
See line 186-187, “40” was changed to “40/52” and “46” was changed to “46/52.”
Reviewer 2 Report
Comments and Suggestions for Authors
The manuscript by Cafariello et al., describes an evaluation of the use of spleen swab samples for the detection of pigs infected with African swine fever virus (ASFV). ASF is a major threat to the pig production industry globally. There are well-established real time quantitative PCR (qPCR) assays for the detection of the viral genome and this manuscript uses such assays to evaluate whether taking swab samples from the spleens of dead pigs is a more efficient means of screening large numbers of pigs for the presence of ASFV than using spleen homogenates which take more time to prepare. The authors demonstrate that the spleen swab samples are at least as good as spleen homogenates. However, it is not clear that this system is better than the use of other samples for this purpose. There are also some important concerns about the processing of some of the data, Ct values are not “normal” numbers and should not be treated as such.
Specific points:
1) Figure 1 adds very little and can be deleted.
2) The authors have calculated mean values (+/- SD? (not stated)) for the Ct values (see lines 160-161, 167, 169, 202 ). It is essential to be aware that Ct values are essentially on a log2 scale thus Ct values of 20 and 30 do not have a mean value of 25 (as if they are “normal” numbers) but about 21. A Ct value of 20 corresponds to a level of the target DNA some 1024-(2^10) times higher than a Ct of 30. Thus, the simple averaging of the Ct values in Table 1 that are between 38.27 and 16.38 (for example, for the homogenates in the Tignon assay) is wrong. This also means that the S.D. values are wrong. To average such numbers, it is first necessary to convert them to the number of genome copies from a standard curve. However, it should be considered how meaningful the average of such numbers differing by more than 2^20-fold actually is. These comments also apply to the data presented in Supplementary Table 2 where means are calculated and for the evaluation of statistical significance.
3) It seems unfortunate that the authors did not perform a more exhaustive comparison of the use of alternative samples that can be collected easily from either living or dead pigs (e.g. use of nasal or oral swabs, use of blood or serum or meat juice) so that more useful conclusions could be made from such studies.
Minor points
a) There is inconsistency about the use of ASFV (the virus) and ASF (the disease), the authors should check which they mean (e.g. see lines 20, 38).
b) The authors should be consistent in using “µl” and not “ul” (e.g. line 128, 192)
Comments on the Quality of English LanguageGenerally the use of English is OK but some minor errors are apparent, a bit of a surprise for a manuscript from an English speaking country.
Author Response
Comment 1.
The manuscript by Cafariello et al., describes an evaluation of the use of spleen swab samples for the detection of pigs infected with African swine fever virus (ASFV). ASF is a major threat to the pig production industry globally. There are well-established real time quantitative PCR (qPCR) assays for the detection of the viral genome and this manuscript uses such assays to evaluate whether taking swab samples from the spleens of dead pigs is a more efficient means of screening large numbers of pigs for the presence of ASFV than using spleen homogenates which take more time to prepare. The authors demonstrate that the spleen swab samples are at least as good as spleen homogenates. However, it is not clear that this system is better than the use of other samples for this purpose. There are also some important concerns about the processing of some of the data, Ct values are not “normal” numbers and should not be treated as such.
Response: We would like to thank the reviewer for their valuable and critical comments.
Specific points:
1) Figure 1 adds very little and can be deleted.
Response: While it is understandable that the reviewer feels that Figure 1 could be omitted; the goal of this publication is ultimately to broadly assist animal health laboratories across Canada and globally to increase ASF surveillance throughput and maintain screening capacity in the event of an outbreak. We believe that the inclusion of Figure 1 and especially panel C of Figure 1 should be maintained as it provides laboratory analysts with a visual representation of the spleen swab preparation procedure upon receiving spleen tissues to the lab. As well, this visual resource could become highly valuable to analysts in countries where English is not a native language.
2) The authors have calculated mean values (+/- SD? (not stated)) for the Ct values (see lines 160-161, 167, 169, 202). It is essential to be aware that Ct values are essentially on a log2 scale thus Ct values of 20 and 30 do not have a mean value of 25 (as if they are “normal” numbers) but about 21. A Ct value of 20 corresponds to a level of the target DNA some 1024-(2^10) times higher than a Ct of 30. Thus, the simple averaging of the Ct values in Table 1 that are between 38.27 and 16.38 (for example, for the homogenates in the Tignon assay) is wrong. This also means that the S.D. values are wrong. To average such numbers, it is first necessary to convert them to the number of genome copies from a standard curve. However, it should be considered how meaningful the average of such numbers differing by more than 2^20-fold actually is. These comments also apply to the data presented in Supplementary Table 2 where means are calculated and for the evaluation of statistical significance.
Response: We particularly thank the reviewer for this critical feedback on the data analysis and Ct values. We have removed the mean and standard deviation values used in the manuscript for the Tignon, Zsak and Moniwa assays. Please see the following highlighted lines in the revised manuscript (lines 179-189).
In fact, the mean and standard deviation values do not contribute as much to the stated conclusion – that spleen swabbing is a better strategy for sample preparation than tissue homogenization – as do the % positivity and correlational data from both table 1 and figure 2 respectively. Since we have not included a standard curve for ASFV genome or β-Actin real-time PCR assays upon running these tests, it would be impossible for us to convert these Ct values to copy number without applying a theoretical regression (y = 3.32x + 40) which we know is not accurate from previous experience with these assays.
For the same reason, we cannot include copy number information for the Armored RNA Enterovirus data discussed in line 233 or Table S2. In this case we have decided to leave the mean and standard deviation figures in the manuscript as we believe it is important to note that they are in fact comparable while we have omitted any conclusion confirming that PCR inhibition is not present.
As admitted in line 259-261 the reason for the increase in sensitivity upon swabbing is not completely understood. Moreover, we have removed the t-test calculations from the Table S2 since this statistical calculation is likely not valid.
3) It seems unfortunate that the authors did not perform a more exhaustive comparison of the use of alternative samples that can be collected easily from either living or dead pigs (e.g. use of nasal or oral swabs, use of blood or serum or meat juice) so that more useful conclusions could be made from such studies.
Response: We thank the reviewer for their valuable suggestion; however comparison of alternative sample types is outside the scope of this study. In the discussion, we discussed the study conducted in Uganda comparing spleen swabbing and meat juice samples for ASFV detection conducted by Okwasiimire et al., in highlighted lines 271-277 in the discussion of the revised manuscript.
- Okwasiimire, R.; Nassali, A.; Ndoboli, D.; Ekakoro, J.E.; Faburay, B.; Wampande, E.; Havas, K.A. Comparison of diaphragm meat juice and muscle swab samples to spleen and spleen swab samples for the detection of African swine fever viral nucleic acid. J Vet Diagn Invest. 2023, 35(2), 145-152.
Although not compared in the current manuscript, we have conducted numerous experiments and published extensively about various sample types such as nasal, oropharyngeal and rectal swabs, oral fluids, and meat juices for the detection of ASFV genome in a number of previous publications.
- Goonewardene KB, Chung CJ, Goolia M, Blakemore L, Fabian A, Mohamed F, Nfon C, Clavijo A, Dodd KA, Ambagala A. Evaluation of oral fluid as an aggregate sample for early detection of African swine fever virus using four independent pen-based experimental studies. Transbound Emerg Dis. 2021 Sep;68(5):2867-2877. doi: 10.1111/tbed.14175. Epub 2021 Jun 17. PMID: 34075717.
- Onyilagha C, Nash M, Perez O, Goolia M, Clavijo A, Richt JA, Ambagala A. Meat Exudate for Detection of African Swine Fever Virus Genomic Material and Anti-ASFV Antibodies. Viruses. 2021 Sep 1;13(9):1744. doi: 10.3390/v13091744. PMID: 34578325; PMCID: PMC8472811.
Swabs collected from mucosal surfaces (oral, buccal, nasal and rectal) are suitable for live animal sampling when the animals can be properly restrained. However, in terms of sensitivity, mucosal swabs exhibit less sensitivity and inconsistent detection compared to blood. Oral fluids are a suitable aggregate/ group sample type to screen swine herds for the presence of pathogens. Upon detection in oral fluids, confirmation must be done by testing individual blood samples.
World Organization for Animal Health (WOAH) recommends testing blood and tissue samples such as spleen, and certain lymph nodes as the gold standards for the detection of ASF, out of which the tissue samples can only be collected when the animal is dead. Our laboratory tests spleen samples collected from abattoirs and suspected dead pigs for ASF surveillance.
In search of sample types for efficient dead pig surveillance, we recently demonstrated that superficial inguinal lymph nodes are a suitable sample type that can be conveniently collected and tested for the detection of ASFV genome at comparable levels to spleens.
- Goonewardene KB, Onyilagha C, Goolia M, Le VP, Blome S, Ambagala A. Superficial Inguinal Lymph Nodes for Screening Dead Pigs for African Swine Fever. Viruses. 2022 Jan 4;14(1):83. doi: 10.3390/v14010083. PMID: 35062287; PMCID: PMC8780992.
However, despite the ease of collection, even superficial lymph nodes require rigorous tissue homogenization methods. As a result, we were compelled to explore the possibility of testing spleen swabs opposed to spleen tissue homogenates since spleens are already being collected for our ASF surveillance program from slaughterhouses and wild boars; Can Spot ASF.
Minor points
1) There is inconsistency about the use of ASFV (the virus) and ASF (the disease), the authors should check which they mean (e.g. see lines 20, 38).
Response: Upon carrying out real-time PCR for the presence of ASFV genome in an individual sample we are also testing for the presence of ASF disease in a population or geographical area and often both options are appropriate. The reason why “ASF” was originally used in lines 21-21 of the abstract is because the definition of “ASFV” is not provided until the beginning of the introduction (line 39). We have modified line 22 of the abstract to include ASFV in place of ASF.
2) The authors should be consistent in using “µl” and not “ul” (e.g. line 128, 192)
Response: Thank you for this comment. This minor edit has been made.
Reviewer 3 Report
Comments and Suggestions for Authors
This is a very interesting and useful paper that very clearly makes a sound case for the use spleen swab sampling in the detection of African swine fever.
The authors provide a short, but informative, introduction to African swine fever (ASF) with an adequate literature review that illustrates the importance of this significant contagious disease at an international scale.
The authors provide an impressively rigorous and thorough Materials and Methods that describes in great detail the collection and processing of splenic swabs and comparative blood samples and splenic tissue homogenates that enable the experimental establishment of the validity of the more simple and expedient splenic swab collection in the detection of ASW. The informative nature of the Materials and Methods section would enable other researchers to collect samples that are suitable for detection of ASFV genomic DNA using RT-PCr as described in Section 2.3. The practical benefit of this swabbing would be that samples could be collected from wild animals in the field with minimal training of collectors.
Results and Discussion
The practical benefits of swabbing are evident and presented effectively in Table 1. This depicts the RT-PCR results from splenic homogenates and swabs from ASF experimentally infected pigs in three different collection areas. The positive and tight correlations depicted in Fig 2 show that the effectiveness of swabs are very similar to the results from homogenates.
Table 2 showing the results for RTA-PCR comparing homogenates to swabs make another convincing argument for the use of swabs with possible reasons for the ND detection in several of the homogenates where positive results were provided using swabs.
The authors make the quite justified claim that spleen swab samples provide comparable, if not increased sensitivity compared to the use of the spleen tissue homogenates.
The conclusion drawn, which is quite reasonable and justified by the correlations presented, is that the use of spleen swabs is sensitive and more practical to be adopted as a utilitarian method of collection for detection of ASW.
Author Response
Comment 1.
This is a very interesting and useful paper that very clearly makes a sound case for the use spleen swab sampling in the detection of African swine fever.
The authors provide a short, but informative, introduction to African swine fever (ASF) with an adequate literature review that illustrates the importance of this significant contagious disease at an international scale.
The authors provide an impressively rigorous and thorough Materials and Methods that describes in detail the collection and processing of splenic swabs and comparative blood samples and splenic tissue homogenates that enable the experimental establishment of the validity of the more simple and expedient splenic swab collection in the detection of ASW. The informative nature of the Materials and Methods section would enable other researchers to collect samples that are suitable for detection of ASFV genomic DNA using RT-PCr as described in Section 2.3. The practical benefit of this swabbing would be that samples could be collected from wild animals in the field with minimal training of collectors.
Results and Discussion
The practical benefits of swabbing are evident and presented effectively in Table 1. This depicts the RT-PCR results from splenic homogenates and swabs from ASF experimentally infected pigs in three different collection areas. The positive and tight correlations depicted in Fig 2 show that the effectiveness of swabs are very similar to the results from homogenates.
Table 2 showing the results for RTA-PCR comparing homogenates to swabs make another convincing argument for the use of swabs with possible reasons for the ND detection in several of the homogenates where positive results were provided using swabs.
The authors make the quite justified claim that spleen swab samples provide comparable, if not increased sensitivity compared to the use of the spleen tissue homogenates.
The conclusion drawn, which is quite reasonable and justified by the correlations presented, is that the use of spleen swabs is sensitive and more practical to be adopted as a utilitarian method of collection for detection of ASW.
Response: We thank the reviewer for the comments.
Reviewer 4 Report
Comments and Suggestions for Authors
This study is well designed and the manuscript accurately describes the methods, results and outcomes of the study. They have added an important data set to research conducted by others for the use of swabs in place of homogenisation of tissues, in this instance for another important virus. One benefit that they could consider is that collection of swabs from tissues during post-mortem examinations under field conditions makes both sample collection and transport much easier. Swabs are readily collected and placed in a suitable liquid virus transport medium (VTM) that avoids the need to collect and package tissue samples. Experience has shown that nucleic acid can remain stable in a good quality VTM at 4oC for many weeks. For situations where virus isolation is not needed there are commercially available VTM solutions such as "PrimeStore" that have proven capacity to fully inactivate high risk viruses such as ASFV, providing both nucleic acid stability and removing the need to ship material under special transport conditions. This suggestion is not an essential inclusion but a short paragraph would enhance a nice study that is well written even further
Author Response
Reviewer #4.
Comment: This study is well designed and the manuscript accurately describes the methods, results and outcomes of the study. They have added an important data set to research conducted by others for the use of swabs in place of homogenisation of tissues, in this instance for another important virus. One benefit that they could consider is that collection of swabs from tissues during post-mortem examinations under field conditions makes both sample collection and transport much easier. Swabs are readily collected and placed in a suitable liquid virus transport medium (VTM) that avoids the need to collect and package tissue samples. Experience has shown that nucleic acid can remain stable in a good quality VTM at 4oC for many weeks. For situations where virus isolation is not needed there are commercially available VTM solutions such as "PrimeStore" that have proven capacity to fully inactivate high risk viruses such as ASFV, providing both nucleic acid stability and removing the need to ship material under special transport conditions. This suggestion is not an essential inclusion but a short paragraph would enhance a nice study that is well written even further.
Response: We thank the reviewer for the comments and the great ideas that would certainly enhance our study. The present study was conducted using archived spleen samples collected during different animal experiments and the spleens that were submitted to NCFAD as part of the ongoing CanSpot ASF surveillance program. As a result, spleen swabs in the present study were collected in the laboratory. It would be convenient to collect spleen swabs directly during postmortem examination and send to the laboratory instead of sending whole tissue samples. However, it is also important to ensure that the spleen swab is collected in a uniform manner with maximum representation of the spleen by all veterinarians. For the time being, it is preferred that the collection of spleen swab is done in the laboratory in a consistent manner.
We are currently carrying out our own studies related to the use of both VTM and Primestore reagents in our ASF work. As well, we have also tested the use of GeneBio Systems Inc. SkinnyTubes for the purpose of producing spleen swab samples for ASF detection in the field. We have not mentioned these reagents and tools in the present manuscript primarily for the purpose of brevity. Our work with these reagents and tools will be published in near future.
Round 2
Reviewer 2 Report
Comments and Suggestions for Authors
The revised manuscript by Cafariello et al., has been improved but there remain some issues that are inadequately addressed.
1) I still believe Figure 1 adds very little and should be omitted from the main text. If wished, the authors could move it to Supplementary Information.
2) The authors have now removed the inappropriately calculated mean and SD values for the Ct values from the ASFV qPCR assays. However, they still use this approach for the Armored RNA assay, see lines 228-231 and Table S2. This also has to be changed, it is just incorrect. The authors supply all the individual data points in Table S2, thus the incorrectly calculated mean and S.D. values should just be removed from the Table and the text. It is illogical to remove such values from the ASFV data set but not from the Armored RNA data.
Author Response
Comment 1. I still believe Figure 1 adds very little and should be omitted from the main text. If wished, the authors could move it to Supplementary Information.
Response: This figure was moved to the supplemental materials and named “Figure S1”, all other figures were adjusted as required. Please see lines 102, 232, and the supplemental materials.
Comment 2: The authors have now removed the inappropriately calculated mean and SD values for the Ct values from the ASFV qPCR assays. However, they still use this approach for the Armored RNA assay, see lines 228-231 and Table S2. This also has to be changed, it is just incorrect. The authors supply all the individual data points in Table S2, thus the incorrectly calculated mean and S.D. values should just be removed from the Table and the text. It is illogical to remove such values from the ASFV data set but not from the Armored RNA data.
Response: The mean and standard deviations associated with the Armored Entero RNA assay has been removed from both the manuscript as well as the supplemental materials. Please see line 219 and the supplemental materials.